# Exposure to Obesogenic Environments during Perinatal Development Modulates Offspring Energy Balance Pathways in Adipose Tissue and Liver of Rodent Models

**DOI:** 10.3390/nu15051281

**Published:** 2023-03-04

**Authors:** Diana Sousa, Mariana Rocha, Andreia Amaro, Marcos Divino Ferreira-Junior, Keilah Valéria Naves Cavalcante, Tamaeh Monteiro-Alfredo, Cátia Barra, Daniela Rosendo-Silva, Lucas Paulo Jacinto Saavedra, José Magalhães, Armando Caseiro, Paulo Cezar de Freitas Mathias, Susana P. Pereira, Paulo J. Oliveira, Rodrigo Mello Gomes, Paulo Matafome

**Affiliations:** 1Coimbra Institute for Clinical and Biomedical Research (iCBR) and Institute of Physiology, Faculty of Medicine, University of Coimbra, 3000-548 Coimbra, Portugal; 2Center for Innovative Biomedicine and Biotechnology (CIBB), University of Coimbra, 3000-548 Coimbra, Portugal; 3Clinical Academic Center of Coimbra (CACC), 3000-061 Coimbra, Portugal; 4Department of Physiological Sciences, Institute of Biological Sciences, University Federal of Goiás, Goiânia 74690-900, Brazil; 5Hospitals of the University of Coimbra, 3004-561 Coimbra, Portugal; 6Laboratory of Secretion Cell Biology, Department of Biotechnology, Genetics and Cell Biology, State University of Maringa, Maringa 87020-900, Brazil; 7Laboratory of Metabolism and Exercise (LaMetEx), Research Centre in Physical Activity, Health and Leisure (CIAFEL) and Laboratory for Integrative and Translational Research in Population Health (ITR), Faculty of Sports, University of Porto, 4200-450 Porto, Portugal; 8Polytechnic Institute of Coimbra, Coimbra Health School (ESTeSC), 3046-854 Coimbra, Portugal; 9Molecular Physical-Chemistry R&D Unit, Department of Chemistry, University of Coimbra, 3004-535 Coimbra, Portugal; 10CNC—Centre for Neuroscience and Cell Biology, University of Coimbra, 3004-504 Coimbra, Portugal

**Keywords:** metabolic diseases, energy balance, metabolic programming, sugars and AGEs, obesity/adipose tissue

## Abstract

Obesogenic environments such as Westernized diets, overnutrition, and exposure to glycation during gestation and lactation can alter peripheral neuroendocrine factors in offspring, predisposing for metabolic diseases in adulthood. Thus, we hypothesized that exposure to obesogenic environments during the perinatal period reprograms offspring energy balance mechanisms. Four rat obesogenic models were studied: maternal diet-induced obesity (DIO); early-life obesity induced by postnatal overfeeding; maternal glycation; and postnatal overfeeding combined with maternal glycation. Metabolic parameters, energy expenditure, and storage pathways in visceral adipose tissue (VAT) and the liver were analyzed. Maternal DIO increased VAT lipogenic [NPY receptor-1 (NPY1R), NPY receptor-2 (NPY2R), and ghrelin receptor], but also lipolytic/catabolic mechanisms [dopamine-1 receptor (D1R) and p-AMP-activated protein kinase (AMPK)] in male offspring, while reducing NPY1R in females. Postnatally overfed male animals only exhibited higher NPY2R levels in VAT, while females also presented NPY1R and NPY2R downregulation. Maternal glycation reduces VAT expandability by decreasing NPY2R in overfed animals. Regarding the liver, D1R was decreased in all obesogenic models, while overfeeding induced fat accumulation in both sexes and glycation the inflammatory infiltration. The VAT response to maternal DIO and overfeeding showed a sexual dysmorphism, and exposure to glycotoxins led to a thin-outside-fat-inside phenotype in overfeeding conditions and impaired energy balance, increasing the metabolic risk in adulthood.

## 1. Introduction

Since the 1980s, the incidence and prevalence of obesity and type 2 diabetes (T2D) have been escalating worldwide, being associated with Westernized diet intake and a sedentary lifestyle [1,2,3]. Abnormal body fat accumulation in obesity can promote the development of other diseases such as T2D, characterized by pancreatic β-cells dysfunction and insulin resistance in target organs [4,5]. Insulin resistance in peripheral organs stimulates insulin’s continuous release, leading to hyperinsulinemia and the exhaustion of pancreatic β-cells [4,6,7]. Furthermore, increased levels of free fatty acids (FFAs) in obesity contribute to lipotoxicity, one of the main factors for insulin resistance [8]. Overfeeding and energy balance dysregulation are among the main factors contributing to the development of obesity [9]. Under obesogenic conditions, energy balance regulators such as ghrelin, neuropeptide Y (NPY), leptin, glucagon-like peptide-1 (GLP-1), and dopamine levels are altered, disrupting the mechanisms involved with insulin secretion and energy storage and expenditure [10,11,12,13,14]. Furthermore, westernized diets are rich in saturated and monounsaturated fats, simple carbohydrates, and poor in fibers, making them a major source of advanced glycation end-products (AGEs) and their precursors—glycotoxins (reviewed by [15,16,17,18,19]). Previous reports from our laboratory showed that adipose tissue glycation in high-fat diet-fed rats impairs its expandability—which may be related to energy balance mechanisms dysregulation—leading to insulin resistance [20,21]. Moreover, glycation reduction through pharmacological strategies prevents such harmful effects [21,22].

Neuroendocrine pathways such as NPY and dopamine act in peripheral tissues to regulate lipid and glucose metabolism [23,24,25]. However, the effects mediated by dopamine and NPY depend on their receptor subtype. Different dopamine receptors trigger opposite effects: while dopamine receptor 1 (D1R) induces lipolysis and catabolic activity through AMP-activated protein kinase (AMPK) activation [26,27], dopamine receptor 2 (D2R) inhibits lipolysis by decreasing AMPK activity, hormone-sensitive lipase (HSL), and ATP citrate lyase (ACL), and it also induces lipogenesis by increasing Acetyl-CoA carboxylase (ACC) activity [25,26]. Regarding NPY receptors in white adipose tissue (WAT), NPY receptor 1 (NPY1R) induces lipogenic effects, while NPY receptor 2 (NPY2R) is associated with adipogenic and angiogenic processes [23,28,29,30,31]. Known as the stomach-derived hunger hormone, ghrelin binding to growth hormone secretagogue receptor 1α (GHS-R1α) in the hypothalamus regulates energy balance by NPY/Agouti-related protein (AgRP) neurons activation [32,33]. In WAT, the direct activation of GHS-R1α on adipocytes decreases insulin sensitivity and stimulates adiposity [12,34,35]. Acyl-ghrelin in retroperitoneal adipose tissue (AT) increases sterol regulatory element-binding transcription factor 1 (SREBP1C), a master regulator of lipogenesis, while decreasing fatty acid (FA) transport, which contributes to fat accumulation [35,36]. Ghrelin acts as an anti-inflammatory agent in the liver and promotes hepatic lipogenesis by activating the mTOR-PPARy signaling pathway, although its role on glucose and lipid metabolism remains unknown [34,37]. Overall, both NPY and acyl ghrelin levels are increased in patients with obesity and T2D, contributing to adiposity and reduced insulin sensitivity [11,12,38,39], whereas dopamine action is dependent on subtype receptors.

Maternal nutrition impacts offspring gene expression and epigenome, metabolism, and cellular function, affecting organ development and later newborns’ lives [40]. Gestation and lactation play a major role in programming windows. Maternal nutrition and metabolic status induce alterations in the intrauterine environment and breastmilk composition that determine offspring adiposity [41,42]. Worldwide, studies demonstrated a higher risk of obesity development when the organism was exposed to maternal overnutrition and obesity [43,44,45]. At the central level, it has already been demonstrated that both maternal obesity and postnatal overfeeding, besides leading to overweight, induce NPY hypothalamic changes in offspring [40,46]. Moreover, maternal obesogenic diets, such as westernized diets, can induce obesity and contribute to metabolic complications during lactation [47,48], showing the importance of maternal diet/lifestyle during lactation. Thus, it is necessary to understand if insulin-sensitive tissues such as the liver and AT undergo alterations in the pathways that regulate energy expenditure after exposure to an unhealthy maternal lifestyle and postnatal overfeeding.

In the present study, we compared the role of different obesogenic environments, namely maternal hypercaloric diets (gestation and lactation) and postnatal overfeeding, in energy balance mechanisms, particularly ghrelin, NPY, and dopamine signaling in insulin-sensitive tissues of young animals. Additionality, in order to disclose the role of maternal diet in the metabolic state, we addressed the impact of glycotoxins, common in Western diets, in postnatal overfed rats. We hypothesize that animals with postnatal overweight induced by both maternal obesity and overfeeding show an adaptation of the mechanisms of energy storage and expenditure to unhealthy motherhood and the larger amount of food available. However, exposure to maternal glycation in postnatal overfed rats may hamper these compensatory mechanisms, aggravating the risk for obesity, T2D, and other metabolic diseases that are major healthcare concerns worldwide.

## 2. Materials and Methods

### 2.1. In Vivo Models of Obesogenic Metabolic Programming

#### 2.1.1. Maternal Diet-Induced Obesity (DIO) during Gestation and Lactation

Female Sprague-Dawley rats were fed an HFHS diet (containing 42% metabolizable energy from fat, 27% from proteins, and 31% from carbohydrates) before pregnancy until lactation. At PND 21, newborns were weaned and fed with chow. At PND 42, male and female offspring were euthanized, and peripheral tissues were collected for molecular and cellular analysis. In this study, liver and visceral AT (VAT) samples were used from a previously published study, where the experimental design (Figure 1A) and all the biochemical profiles of the rats with 42 days were already described (control litters = 6; HFHS litters = 6; number of males from control dams = 4; number of males from dams submitted to HFHS diet = 5; number of females from control dams = 4; number of females from dams submitted to HFHS diet = 5). More information about this animal model is available in Stevanović-Silva et al. [49,50]. The results of this animal model are depicted in Figure 1 and Figure 2. 

#### 2.1.2. Postnatal Overfeeding and Glycation Models

The procedures were approved by the Animal Welfare Committee (ORBEA) of the Coimbra Institute for Clinical and Biomedical Research (iCBR), Faculty of Medicine, University of Coimbra. Animal experimentation was performed following the European Community directive guidelines for the use of laboratory animals (2010/63/EU), transposed into Portuguese law in 2013 (Decreto-Lei 113/2013). Wistar rats were housed under standard conditions (ventilation; 22 °C temperature; 55% humidity; 12 h/12 h light/dark cycle) with ad libitum access to food and water. This work used three animal models of young Wistar rats (postnatal overfeeding model, a maternal glycation model, and postnatal overfed rats exposed to Maternal Glycotoxins). After delivery, all litter sizes were reduced to 8 pups for standardization. After birth, body weight was monitored at postnatal day (PND) 0, PND 4, PND 7, PND 14, PND 21, PND 35, and PND 45. On PND 21, newborns were weaned and separated from their mothers until PND 45 and fed a standard diet. During this period, food consumption was weekly monitored. At PND 45, triglyceride levels were measured in the tail vein (Accutrend, Roche, Mannheim, Germany), and an insulin tolerance test was performed. After blood collection, animals were anesthetized with an IP injection of ketamine/chlorpromazine and euthanized by cervical displacement, and the VAT and liver were collected for molecular analysis. The dams were anesthetized with an IP injection of ketamine/chlorpromazine, milk samples were collected, and the females were euthanized by cervical displacement at 21 PND for liver collection and tissue morphology analyses. 

##### Postnatal Overfeeding Model

On the third day after the birth, a small litter (SL) protocol was implemented by reducing the litter size to 3 pups per dam to induce postnatal overfeeding and overweight (Figure 3A) (control litters = 7; SL litters = 5; number of males from normal litters (NL) = 35; number of males from SL = 9; number of females from NL = 12; number of females from SL = 6). The results of this animal model are depicted in Figure 3 and Figure 4.

##### Maternal Glycation Model

Wistar dams were injected via intraperitoneal (IP) with S-P-Bromobenzylgutathione cyclopentyl diester—BBGC (5 mg/kg)—a selective inhibitor of Glyoxalase 1 (GLO1), during the first six days post-partum, whereas vehicle dams were injected with the vehicle dimethyl sulfoxide—DMSO (60 µL) (Figure 5A) (control litters = 7; vehicle litters = 5; BBGC group = 5; number of male pups from control dams = 35; number of male pups from dams treated with vehicle = 22; number of male pups from dams treated with BBGC = 29; number of female pups from control dams = 12; number of female pups from dams treated with vehicle = 9; number of female pups from dams treated with BBGC = 9). The results of this animal model are depicted in Figure 5, Figure 6 and Figure 7.

##### Postnatal Overfed Rats Exposed to Maternal Glycotoxins Model

Male offspring from dams treated with BBGC as described in the maternal glycation model were submitted to an SL reduction to 3 pups for litter at PND 3 (Figure 8A) (SL litters = 5; BBGC + SL litters = 3; number of SL = 9; number of BBGC + SL = 9). The results of this animal model are depicted in Figure 8.

### 2.2. Milk Sample Collection and Determination of Total Antioxidant Capacity and Triglycerides

The Wistar female dams were anesthetized and injected with oxytocin (Facilpart) at a concentration of 10 UI/mL after 6 h of fasting, on day 21 postpartum. Milk samples were collected, milk triglycerides were determined using the Accutrend, Roche, Germany, and milk total antioxidant capacity was assessed with an assay kit (ab65329) according to the manufacturer’s instructions.

### 2.3. Plasma Determinations

Wistar rats’ blood samples were collected by cardiac puncture under anesthesia and immediately before sacrifice in Vacuette K3EDTA tubes (Greiner Bio-one, Kremsmunster, Austria) at PND 45. Blood samples were immediately centrifuged (2200× *g*, 4 °C, 15′) and the plasma fraction was stored at −80 °C until performing the Rat Insulin ELISA Kit (Mercodia, Uppsala, Sweden), according to the manufacturer’s instructions. Total and HDL cholesterol were determined using the Prestige 24i Tokyo Boeky system with reagents from Cormay, Poland.

### 2.4. Histology—Haematoxylin-Eosin

Livers and VAT from dams and offspring of exclusive metabolic programming during lactation models were fixed in formalin solution (10%), dehydrated in an increasing series of alcohol concentrations (70% to 100%), cleared in xylene, and then embedded in histological paraffin. The livers were sectioned in a microtome, on a non-serial section of 4 µm thickness (n = 3/group) and subsequently dried overnight at room temperature (RT). The paraffin-embedded liver sections were submitted to paraffin-removing protocols, using xylol, progressive hydration (EtOH 100%/70%/30% 3′/each and Milli-Q water during 3′ at RT), and stained with hematoxylin and eosin (H&E). Then, the liver sections were washed again, and coverslips were mounted using a mounting medium (DAKO, Kyoto, JAPAN). Lastly, images (100×) were captured in a Zeiss microscope with an incorporated camera (Zeiss, Jena, Germany). All full-size representative images in the manuscript are presented in the Appendix A.

### 2.5. Western Blot

Hepatic and VAT samples were collected and washed with PBS and disrupted in lysis buffer (0.25 M Tris-HCl, 125 mM NaCl, 1% TritonX-100, 0.5% SDS, 1 mM EGTA, 1 mM EDTA, 20 mM NaF, 2 mM Na3VO4, 10 mM βglycerophosphate, 2.5 mM sodium pyrophosphate, 10 mM PMSF, 40 µL of protease inhibitor) using the TissueLyser system (Quiagen, Hilden, Germany). The bicinchoninic acid (BCA) Protein Assay Kit was carried out on the supernatant (14,000 rpm for 20 min at 4 °C, followed by the addition of Laemmeli buffer (62.5 mM Tris-HCl, 10% glycerol, 2% SDS, 5% β-mercaptoethanol, and 0.01% bromophenol blue). Tissue samples (20 µg) were loaded onto SDS-PAGE and electroblotted into polyvinylidene difluoride (PVDF) membrane (Advansta, San Jose, CA, USA). Tris-buffered saline-tween (TBS-T) 0.01% and bovine serum albumin (BSA) 5% were used to block the membranes, which were then incubated with primary (overnight, 4 °C) and secondary antibodies (2 h RT), following the dilutions listed in the Appendix A. The proteins of interest were detected using enhanced chemiluminescence (ECL) substrate with the LAS 500 system (GE Healthcare, Chicago, IL, USA). The bands of interest were quantified with Image Quant 5.0 software (Molecular Dynamics). The results were expressed as a percentage of control and normalized for the loading control (calnexin, 83 kDa).

### 2.6. Statistical Analyses

The results are presented as the mean ± standard error of the mean (SEM). Statistical analysis was performed with GraphPad Prism 8 (GraphPad Software, Inc., San Diego, CA, USA). The normality of the data was assessed with the Shapiro–Wilk normality test. Accordingly, data with two conditions were analyzed with a nonpaired *t*-test or Mann–Whitney test, and data with more than two conditions were analyzed with the Kruskal–Wallis test or with a one-way ANOVA followed by Tukey’s post hoc test. Differences were considered for *p* < 0.05.

## 3. Results

### 3.1. Liver and VAT Sexual Dysmorphism in the Response to Different Perinatal Obesogenic Environments

#### 3.1.1. Maternal DIO Modulates Energy Balance Mechanisms in the VAT and Liver of Male Offspring

Exposure to maternal obesity increased male offspring body weight until weaning day, as previously published by Stevanović-Silva et al. [50] (summary presented in Figure 1B). Maternal obesity did not cause alterations in hepatic levels of IR (Figure 1C), AMPK (Figure 1D), phosphorylated AMPK (Figure 1E), and PPARα in male offspring (Figure 1F). Exposure to the maternal high-fat high-sugar (HFHS) diet induced a drastic reduction in D1R levels in the male offspring liver (*p* < 0.001 vs. control) (Figure 1H) without affecting the levels of NPY1R (Figure 1G).

Regarding VAT, maternal obesity induced by the HFHS diet decreased total IR levels (*p* < 0.05 vs. control) (Figure 1I) in male offspring without changing AMPK (Figure 1J) or PPARγ levels (Figure 1L). NPY1R (Figure 1M), NPY2R (Figure 1N), and GHS-R1α (Figure 1O) were significantly increased in male offspring VAT (*p* < 0.001 vs. control, *p* < 0.05 vs. control, and *p* < 0.05 vs. control, respectively), suggesting adipogenesis and lipogenesis upregulation. Moreover, the maternal HFHS diet increased male offspring D1R levels (*p* < 0.05 vs. control) (Figure 1P) and AMPK phosphorylation (*p* < 0.05 vs. control) (Figure 1K) in VAT compared to offspring from dams fed a standard diet.

#### 3.1.2. Maternal Obesity Alters Levels of Energy Balance-Regulating Receptors in the VAT of Female Offspring

Maternal DIO female offspring were also studied to assess a potential sexual dimorphism in the liver and VAT adaptation to the maternal metabolic state. As occurs in males, maternal obesity increased female offspring body weight until weaning day (in summary in Figure 2A) (previously published [49]). Similar to male livers, there were no alterations in total IR levels (Figure 2B), and a significant reduction in D1R (Figure 2G) in female offspring exposed to the maternal HFHS diet during the perinatal period was observed (*p* < 0.001 vs. control). Nevertheless, in female offspring livers, NPY1R, PPARα, and p-AMPK levels also decreased (*p* < 0.05 vs. control, *p* < 0.05 vs. control, and *p* < 0.05 vs. control, respectively) (Figure 2D–F, respectively), while total levels of AMPK increased (*p* < 0.05 vs. control) (Figure 2C), showcasing the disparity between sexes.

Regarding VAT, besides the increased p-AMPK in female offspring (*p* < 0.05 vs. control) (Figure 2J), the total levels of AMPK also augmented (*p* < 0.05 vs. control) (Figure 2I), suggesting that the higher activation of AMPK is a consequence of additional AMPK being available. The changes in the NPY2R, D1R, and total IR levels observed in males were not noticed in the VAT of female offspring (Figure 2H,M,O). Surprisingly, the NPY1R levels were significantly decreased in the female offspring VAT (*p* < 0.05 vs. control) (Figure 2L), demonstrating the sexual dysmorphism of such mechanisms. In the case of the ghrelin receptor, as well as in males, an increase in VAT was observed in female offspring (*p* < 0.05 vs. control) (Figure 2N).

#### 3.1.3. Postnatal Overfeeding Modulates Energy Balance Mechanisms in the VAT and Liver of Male Offspring

In this study, we used males from SL as a model of postnatal overfeeding without modification in the maternal diet, shown to induce obesity early in life [46,51]. Indeed, animals from SL had significantly higher body weight than the control group at the weaning PND 21 (*p* < 0.001 vs. control) (Figure 3B). The higher weight gain was maintained over time until the day of euthanasia—PND 45 (*p* < 0.01 vs. control) (Figure 3C), despite no changes in food intake being observed (Figure 3D). The decay of the glucose rate during the insulin tolerance test per minute (kITT) revealed that insulin sensitivity was not affected in SL rats (Figure 3E), although plasma insulin levels were decreased (*p* < 0.001 vs. control) (Figure 3F). Triglyceride (*p* < 0.05 vs. control) (Figure 3G) and HDL cholesterol levels were increased in male overfed rats (*p* < 0.05 vs. control) (Figure 4I) without affecting cholesterol total level (Figure 3H).

Similar to the maternal HFHS diet model (Figure 1), total liver levels of IR (Figure 3J), PPARα (Figure 3M), NPY1R (Figure 3N), AMPK (Figure 3K), and phosphor-AMPK (Figure 3L) were not affected, and D1R levels were decreased (*p* < 0.05 vs. control) (Figure 3O) in the SL animals with higher consumption of breastmilk during the perinatal period. Moreover, histology showed dispersed fat accumulation zones in the liver of overfed male animals (Figure 3P), while no changes were observed in the liver weight (Appendix A).

The alterations observed in VAT in postnatal overweight offspring induced by overfeeding were different from rats exposed to maternal DIO. Interestingly, the total IR levels in the VAT were increased (*p* < 0.01 vs. control) (Figure 3Q) suggesting an adaptation for the reduced levels of insulin in the plasma. AMPK activation was lower in male overfed rats (*p* < 0.001 vs. control) (Figure 3S), while the total AMPK (Figure 3R) and PPARγ (Figure 3T) levels were maintained. Regarding neuroendocrine mechanisms controlling adipogenic and lipogenic processes, NPY1R, GHS-R1α, and D1R levels in WAT were maintained (Figure 3U,W,X, respectively), while an increase in the adipogenic-related NPY2R levels (*p* < 0.05 vs. control) was also observed (Figure 3V). This was consistent with an increase in fat mass in overfed animals (*p* < 0.01) (Figure 3Y), without adipocyte hypertrophy (Figure 3Z,α).

#### 3.1.4. Female Postnatal Overfed Rats Exhibit Similar Metabolic Adaptations to Male Offspring except the Downregulation of VAT Adipogenic Mechanisms

Food intake (Figure 4C) and liver weight (Appendix A) were not altered in SL female animals, and contrary to SL males, postnatal overfeeding did not induce overweight in female rats over the 45 days (Figure 4A,B), nor did it increase triglyceride levels (Figure 4F). Regarding insulin levels, the outcome of postnatal overfeeding was the same in both sexes: insulin plasma levels were reduced (*p* < 0.001 vs. control) (Figure 4E), while insulin sensitivity did not alter according to kITT (Figure 4D). Cholesterol levels in the plasma of overfed females increased (*p* < 0.05 vs. control) (Figure 4G), accompanied by an increase in HDL levels (*p* < 0.05 vs. control) (Figure 4H).

No changes in the studied mechanisms were observed in the postnatally overfed females’ livers: total IR, total AMPK, p-AMPK, PPARα, NPY1R, and D1R levels (Figure 3I–N, respectively). Histology demonstrated few areas of lipid droplet accumulation in the liver (Figure 3O), as occurs in male overfed animals. Regarding AT, total IR levels are maintained in overfed females (Figure 4P), contrary to overfed male rats, as well as total AMPK and p-AMPK levels (Figure 4Q,R, respectively). Regarding energy storage mechanisms, both NPY1R and NPY2R were reduced in the VAT of postnatal overfed females (*p* < 0.01 vs. control and *p* < 0.05 vs. control, respectively) (Figure 4T,U, respectively), which was consistent with decreased PPARγ levels (*p* < 0.01 vs. control) (Figure 4S), unaffected fat mass (Figure 4X) and adipocyte size (Figure 4Y,Z), indicating that lipogenesis and adipogenesis may be reduced in postnatally overfed females. Ghrelin receptor (Figure 4V) and D1R levels were not altered in the VAT of SL females (Figure 4W).

### 3.2. The Role of Maternal Glycation in Impairing the Neuroendocrine Mechanism of Adaptation to Obesogenic Environments

#### 3.2.1. Glycation Changes Breastmilk Composition

The glycation induced by BBGC (5 mg/kg) through IP did not alter the body weight or glycemic profile of dams (Figure 5B,C, respectively). Furthermore, BBGC did not cause toxicity in the liver (Figure 5D). However, glycation changed breastmilk quality, decreasing triglyceride levels (*p* < 0.01 vs. control) and total antioxidant capacity (*p* < 0.05 vs. control) (Figure 5E,F, respectively). A known methylgyoxal-derived AGE is N-(5-hydro-5-methyl-4-imidazolon-2-yl)-ornithine (MG-H1) [52]. This compound was increased in VAT from male offspring exposed to maternal glycotoxins (*p* < 0.05 vs. control; *p* < 0.05 vs. vehicle) (Figure 5G), showing that inhibition of glyoxalse I in dams leads to AGEs accumulation in VAT offspring. 

#### 3.2.2. Exposure to Glycotoxins during Lactation Decreases Offspring Food Intake and Impairs Insulin-Dependent Glucose Uptake without Affecting the Insulin Levels of Male Offspring 

Obesogenic environments contribute to body weight alteration in early life, which is not often observed in lean or adult experimental models of glycation [53]. Here, we observed that maternal glycation did not affect the body weight of male offspring either during the breastfeeding period or after weaning (Figure 6A,B, respectively), although lower food consumption was observed (*p* < 0.01 vs. control) (Figure 6C). Despite no alterations In insulin plasma levels (Figure 6E), the kITT was reduced in male offspring exposed to maternal glycation (*p* < 0.05 vs. control) (Figure 6D), suggesting lower insulin sensitivity. Exposure to glycotoxins reduced triglyceride levels in breastmilk without affecting the plasma triglyceride, total, and HDL cholesterol levels in male offspring (Figure 6F–H).

#### 3.2.3. Exposure to Maternal Glycation Reduces Both NPY and Dopamine Signalling in the Liver from Male Offspring without Affecting the VAT

Although insulin sensitivity was reduced in male offspring exposed to glycotoxins during lactation, total IR levels in the liver and WAT were not altered (Figure 6I,P, respectively). Total and activated AMPK also remained similar between groups in the hepatic tissue (Figure 6J,K, respectively). 

In the liver, maternal glycation induced a reduction in NPY1R (*p* < 0.01 vs. control; *p* < 0.05 vs. vehicle), D1R (*p* < 0.01 vs. control), and PPARα levels (*p* < 0.001 vs. control; *p* < 0.01 vs. vehicle) (Figure 6M,N,L, respectively). Furthermore, maternal glycation appeared to induce portal inflammatory infiltration at the hepatic level (Figure 6O), while no changes were observed in the liver weight (Appendix A).

Adult models exposed to glycated products showed that glycation does not cause significant changes in VAT function in the lean phenotype [53]. Here, we demonstrate that male offspring from dams treated with glyoxalase 1 inhibitor did not present changes in energy expenditure or storage mechanisms in VAT. The receptors of NPY, ghrelin, and dopamine evaluated (NPY1R, NPY2R, GHS-R1α, and D1R, respectively) display preserved levels when compared with the vehicle and with the control (Figure 6T–W, respectively) as well as the downstream proteins AMPK and PPARγ (Figure 6Q–S). Furthermore, neither fat mass (Figure 6X) nor adipocyte size was altered in male offspring exposed to maternal glycation at PND 45 (Figure 6Y,Z).

#### 3.2.4. BBGC-Induced Maternal Glycation Does Not Affect Food intake, Metabolic Profile and VAT Mechanisms of Lipid Storage and Energy Expenditure in Female Offspring

As well as in the male offspring, during the lactation period and after weaning, no alteration in body weight of females exposed to maternal glycation (Figure 7A,B) or liver weight (Appendix A) was observed. Contrary to what was observed in male offspring, food intake was not affected in BBGC female offspring (Figure 7C). Furthermore, no changes in kITT were observed in female offspring (Figure 7D), suggesting that male offspring are more susceptible to alteration in insulin sensitivity and feeding regulation when exposed to the same maternal condition. Although plasma insulin levels in BBGC female offspring were decreased (*p* < 0.01 vs. control), this was also observed in the vehicle group (*p* < 0.01 vs. control) (Figure 7E). Triglyceride (Figure 7F) and HDL (Figure 7H) levels were also unaltered between groups, while cholesterol levels were affected by vehicle (*p* < 0.05 vs. control; *p* < 0.05 vs. BBGC) (Figure 7G). Total IR levels were also maintained in both the liver and WAT of female offspring (Figure 7I,P, respectively). 

The total levels of AMPK were increased in the liver of female offspring exposed to glycotoxins (*p* < 0.001 vs. control; *p* < 0.001 vs. vehicle) (Figure 7J) while its activity remained similar to the other groups (Figure 7K). PPARα levels also remained similar between all conditions (Figure 7L), and no inflammatory infiltration was observed in the livers of female rats exposed to maternal glycation (Figure 7O). NPY1R levels were decreased in females from dams treated with DMSO (*p* < 0.01 vs. control) and from dams subjected to glyoxalase 1 inhibition (*p* < 0.01 vs. control) (Figure 7M). However, the effect on D1R observed in the vehicle group (*p* < 0.01 vs. control) seems to be partially independent of the reduction induced by BBGC (*p* < 0.001 vs. control; *p* < 0.05 vs. vehicle) (Figure 7N), which is in accordance with the results observed in male offspring. 

As occurs in male offspring exposed to maternal glycation during lactation, in BBGC female offspring, the WAT mechanisms associated with energy expenditure and storage were not affected by maternal glycation alone (Figure 7P–W) as were the fat mass and the adipocyte size (Figure 7X–Z). These results demonstrated once again that glycation alone may not be sufficient to alter the energy balance in the VAT of lean animals at PND 45.

#### 3.2.5. Maternal Glycation Inhibits SL-Induced Weight Gain in Male and Impairs VAT Mechanisms of Adaptation to Overfeeding

BBGC also reduced milk triglyceride levels (*p* < 0.05 vs. SL) and total antioxidant capacity (*p* < 0.05 vs. control) in SL dams (Figure 8B,C, respectively). Exposure to maternal glycation during lactation prevented male offspring weight gain induced by postnatal overfeeding from PND14 to PND35 (*p* < 0.05 vs. SL) (Figure 8D,E). No alterations were observed in food intake or the kITT (Figure 8F,G, respectively) or in the liver weight (Appendix A) in the BBGC + SL group as occurs in SL animals with healthy motherhood. The alterations in plasma insulin (*p* < 0.05 vs. control) (Figure 8H) and triglyceride (*p* < 0.05 vs. control) (Figure 8I) levels observed in SL males were maintained in SL + BBGC offspring. Cholesterol levels were reduced in SL rats exposed to maternal glycation (*p* < 0.05 vs. SL) (Figure 8J), while the increase in HDL-cholesterol levels observed in male SL was not noticed when the animals were exposed to glycotoxins (Figure 8K). Total IR levels were maintained in the liver and VAT of male SL + BBGC offspring compared to SL (Figure 8L,S, respectively), and its upregulation in the liver of SL males was maintained in relation to control rats.

AMPK levels and activity in the liver did not alter in BBGC + SL rats (Figure 8M,N) as occurs in animals exposed to both conditions independently. Surprisingly, the PPARα levels were not altered upon exposure to maternal glycation in animals overfed and induced by SL (Figure 8O) as occurred in lean offspring. As well as observed in the liver from lean male offspring exposed to maternal glycation, in obese conditions, the decrease in NPY1R levels was also verified (*p* < 0.01 vs. SL) (Figure 8P), while the downregulation of D1R levels observed with both conditions per se was maintained (*p* < 0.001 vs. control) (Figure 8Q). These changes were associated with liver inflammatory infiltration, which occurs in lean male offspring from dams treated with BBGC (Figure 8R). Importantly, this male offspring did not show visible liver fat accumulation, contrary to the overfed ones. The decrease in p-AMPK levels induced by postnatal overfeeding in VAT was lost (Figure 8U), while the total levels of AMPK were maintained (Figure 8T) in male offspring exposed to glycotoxins. As already mentioned, postnatal overfeeding causes adaptations in VAT, namely NPY2R overexpression. Exposure to glycated products during lactation induced VAT energy balance dysregulation in postnatal overfed offspring. Despite no changes in PPARγ, NPY1R, or GHS-R1α levels in the VAT of overfed offspring from dams treated with BBGC (Figure 8V,W,Y, respectively), NPY2R levels significantly decreased (*p* < 0.05 vs. SL) (Figure 8X) and D1R further decreased compared to the control (*p* < 0.05 vs. control) (Figure 8Z). Interestingly, although the visceral fat mass did not change in these animals (Figure 8α), their adipocyte size was significantly reduced (*p* < 0.05 vs. SL), following reduced NPY2R levels, and several multivesicular adipocytes were observed, a marker of adipocyte dysfunction (Figure 8β,γ).

## 4. Discussion

Exposure to unhealthy motherhood during the perinatal period has been suggested as a risk factor for the rise of metabolic diseases worldwide. Herein, we describe: (1) the distinct metabolic and neuroendocrine effects in the liver and VAT caused by two obesogenic conditions during developmental programming phases, exposure to perinatal maternal HFHS diet and postnatal overfeeding; (2) the sexual dysmorphism of such responses; and (3) the impact of maternal glycation in disrupting mechanisms of adaptation to postnatal overfeeding (Figure 9). 

Maternal gestational obesity induced by the HFHS diet in the same model causes postnatal overweight and hepatic lipid accumulation in male offspring [49,50]. Here we show that this maternal diet decreases liver D1R levels in both sexes. In adult diabetic rats, lower D1R levels were also observed. Its upregulation by bromocriptine was associated with lipid mobilization from the liver and hepatic steatosis improvement [54]. However, Stevanović-Silva et al. showed no lipid accumulation in female offspring [49,50]. Interestingly, the liver of female offspring presented lower levels of NPY1R, which suggests less inhibition of CTP-1 and lipid accumulation [55], regardless of PPARα level reduction.

In male offspring VAT, maternal DIO decreased IR levels, which may impact glucose uptake and triglyceride formation [56]. However, p-AMPK was upregulated, which is known to promote HSL activity and lipid oxidation. This upregulation was associated with increased D1R levels, in accordance with previous reports showing its role in lipid oxidation [27]. Moreover, the receptors associated with adipogenic and lipogenic processes (NPY1R, NPY2R, and GHS-R1α) levels were also increased, suggesting an adiposity effect of maternal DIO in the offspring. These results suggest that maternal obesity may trigger compensatory mechanisms to enhance lipid storage in male offspring VAT while also counteracting this by simultaneously improving energy expenditure in VAT. Such VAT alterations were sex specific since NPY1R levels were decreased in female offspring, suggesting a reduction in lipogenic/adipogenic processes, contrary to what occurs in males. Indeed, fat distribution depends on sex, with females more prone to accumulate subcutaneous AT (SAT), while males store lipids predominantly in VAT [56,57]. SAT has a lower lipolysis rate but is more efficient on the FFAs uptake than VAT [56,57]. On the other hand, VAT has a lower capacity to store lipids and a more inflammatory environment [56,57]. One limitation of our study is the lack of SAT data to compare the same mechanisms in male and female SAT. Thus, the discrepancy between sex may be associated with the fatty liver presented by the male offspring. Males have a larger VAT area than females, exposing the liver to a higher amount of FFA via the portal vein, which induces fat ectopic deposition in the liver [58].

Postnatal overweight was also induced by neonatal overfeeding, although body weight gain did not occur in females and no increased food consumption was observed after weaning. Male overfed rats have lower plasma insulin levels, suggesting insulin release impairment. Accordingly, Robert A. et al. (2002) demonstrated that the SL animal model at two different ages (PND 26 and PND 110) presents an impairment in the release of insulin from β-cells [59]. Interestingly, IR total levels were increased in male VAT, which may be a compensatory response to hypoinsulinemia. In male rats, overfeeding also promoted an impairment of D1R hepatic levels, as occurs in maternal HFHS-induced early-life obesity. These findings suggest that this may be an important mechanism of lipid retention in the liver, being associated with the presence of lipid droplets in this tissue (Figure 3N). In VAT, neonatal overfeeding in male rats also upregulates NPY2R signaling, potentiating AT expansion and preventing hypertrophy, while no other mechanisms were changed. Thus, the impact of postnatal overweight on VAT energy balance mechanisms apparently depends on the nature of the obesogenic environment. In fact, contrarily to rats exposed to maternal DIO, p-AMPK levels were reduced in VAT from overfed male animals. This suggests that, besides higher caloric intake from milk, nutritional cues may have a direct impact on VAT mechanisms of energy expenditure. It is possible that maternal consumption of fats and sugars stimulates not only energy storage but also energy dissipation pathways in the male offspring. Again, sex was shown to be a crucial factor in energy balance mechanisms. Neonatal overfeeding in females reduces lipogenic and adipogenic processes by decreasing NPY/PPARγ signaling, which may be the reason for the absence of weight gain and may be related to the distinct stimulation of different fat pads, as already discussed.

Given that maternal diets are often rich in sugars, we also intended to study the effects of exposure to maternal glycotoxins on the mechanisms of adaptation to obesogenic environments. Previous studies have used direct administration of methylglyoxal (MG)—an AGEs precursor—in dams [48], and we aimed to use a more physiological model using BGGC injection, inhibiting the enzyme responsible for MG detoxification—GLO1. In the study by Francisco et al. (2018), breastmilk triglyceride levels were increased upon maternal MG ingestion. However, we verified the opposite and a lower antioxidant capacity, suggesting that maternal glycation alters breastmilk composition. As described before, AGEs formation debilitates antioxidant defenses (reviewed by [53]), and we have here shown that maternal glycation has also an impact on breastmilk antioxidant capacity and possibly on the detoxification mechanisms of the offspring. As occurs in adults exposed to glycotoxins, BBGC did not affect maternal or male and female offspring body weight [20]. Nevertheless, Francisco et al. (2018) showed that offspring from MG-treated dams present excessive weight gain only after PND 77, which may be a consequence of metabolic dysregulation secondary to the dose used [48]. As observed in adult models, animals exposed to maternal glycation do not develop insulin resistance, glucose dysmetabolism, or changes in insulinemia [20], although a lower kITT indicates an insufficiency in insulin action, suggesting a predisposition to insulin resistance. Regarding female offspring, plasma insulin levels were decreased, although female offspring of DMSO-treated dams also present this deficiency. Thus, there is no evidence of insulin resistance, consistent with several studies that have reported that females are less prone to develop insulin resistance due to hormonal sex differences [57].

Other studies indicate that glycation plays a more prevailing role in obesity than in normal conditions [57]. Here we observed that maternal glycation does not alter the molecular mechanisms associated with energy balance in the WAT of lean female and male offspring. However, we detected that impairment of D1R levels in the livers of both sexes, as observed in the overfeeding models. These results suggest that changes in milk composition induced by glycation also impact the hepatic level, decreasing D1R and PPARα, a transcription factor known to be activated by FA, and thus possibly reducing lipid oxidation [60]. In addition, NPY1R levels are also diminished in the liver of male offspring exposed to maternal glycotoxins. Moreover, maternal glycation induced inflammatory infiltration in the liver of male offspring. Interestingly, NPY1R has been linked with an anti-inflammatory action associated with a pro-inflammatory M1-like phenotype in other regions, namely the kidney and cardiac system [61,62]. Regarding female livers, no major changes were observed besides those associated with the vehicle, DMSO, and the stress of the daily IP injection. Diverse studies have shown that females are more prone to develop obesity when exposed to maternal stress than males [63]. Moreover, the effects of maternal prenatal stress on gut microbiota colonization are sex-dependent [64]. As a regulator of gut hormone release [65], alterations induced by maternal alterations in the microbiota may change the release of gut-derived hormones.

Previous reports from our group showed that weight gain induced by the HF diet in adult Wistar rats is lost when MG is administered, suggesting that glycation impairs the necessary tissue plasticity to respond to higher nutritional fluxes [20,53]. Indeed, the weight gain provoked by postnatal overfeeding is lost when the young animals are exposed to maternal glycotoxins during lactation. However, the metabolic effects of overfeeding in male offspring regarding plasma triglycerides and insulin levels were maintained when exposed to maternal glycation. Such inhibition of weight gain was associated with the loss of compensatory NPY2R upregulation in overfeeding-induced overweight rats, suggesting a decrease in adipogenesis rates promoting a lower energy storage capacity. Indeed, despite the fact that maternal glycation did not change fat mass, the adipocytes were smaller when the animals experienced maternal glycation. Moreover, several regions show multivesicular adipocytes, a marker of impaired lipid storage and adipocyte dysfunction. Thus, maternal glycation may modify the AT environment, especially in conditions of overfeeding/WAT expansion. So, exposure to glycotoxins during lactation in obesogenic conditions apparently inhibits the development of storage mechanisms to compensate for higher food consumption. These alterations predispose to the development of metabolic syndrome due to their contribution to adipocyte hypertrophy, lipotoxicity, hypoxia in the AT, and insulin resistance, hallmarks of an unhealthy WAT phenotype.

In the liver, maternal glycation impairs NPY1R in lean and overfed animals. Moreover, the hepatic inflammatory infiltration provoked by glycotoxins exposure during lactation was also noticed. These results suggest that alterations in NPY1R signaling and inflammatory infiltration are an exclusive consequence of maternal glycation exposure, as in the livers of overfed animals, NPY1R levels were preserved.

## 5. Conclusions

Dysregulation of energy balance mechanisms is one of the major factors contributing to the development of metabolic diseases. Insults such as obesogenic environments during the perinatal period program adaptations of nutrient-sensing mechanisms, readjusting anabolic/catabolic processes. Here, we characterized, for the first time to our knowledge, the changes in VAT and liver energy balance pathways caused by maternal obesity during pregnancy and postnatal overfeeding, although different rat strains were used. Besides the weight gain, young male animals exposed to the maternal HFHS diet developed compensatory mechanisms regarding energy storage and expenditure, allowing a higher AT plasticity. Postnatal overfeeding apparently protects from AT dysfunction by enhancing energy storage. So, the different consequences in both models suggest that maternal nutritional cues during lactation have a crucial role in their modulation. Moreover, females showed distinct mechanisms in both models, suggesting that lipid partitioning may be different in both sexes since the early stages of development and the energy balance mechanisms in SAT should be of future interest. Nevertheless, despite developing these compensatory peripheral mechanisms, modulation of such pathways can become unhealthy since these compensatory actions may be lost later in life. This may disturb energy balance and therefore contribute to predisposing to the development of metabolic complications. 

To characterize the role of Westernized diet consumption during gestation and lactation, we studied the role of maternal glycation under normal and obese conditions on VAT and liver energy balance mechanisms. Although glycated products do not have major effects on a lean phenotype, they are related to a compromise of AT plasticity, which may lead to AT dysfunction and lipotoxicity when combined with an obesogenic environment (overfeeding). We followed a protocol based on GLO-1 inhibition in order to avoid possible supraphysiological doses after MG administration. However, it is possible that exposure to MG in high-sugar diets may be even higher than in our protocol. In the future, both protocols may be compared for such effects.

Our work points out the relevance of better characterizing these mechanisms in later stages of development and how they predispose to insulin resistance and metabolic diseases at a more advanced age. The role played by pregnancy and lactation should also be disclosed using cross-fostering. Another concern in the future is understanding the importance of offspring environment upon unhealthy motherhood and how the offspring can change its fate and decrease the predisposition of metabolic disease development to break the intergenerational cycle of obesity.

## Figures and Tables

**Figure 1 nutrients-15-01281-f001:**
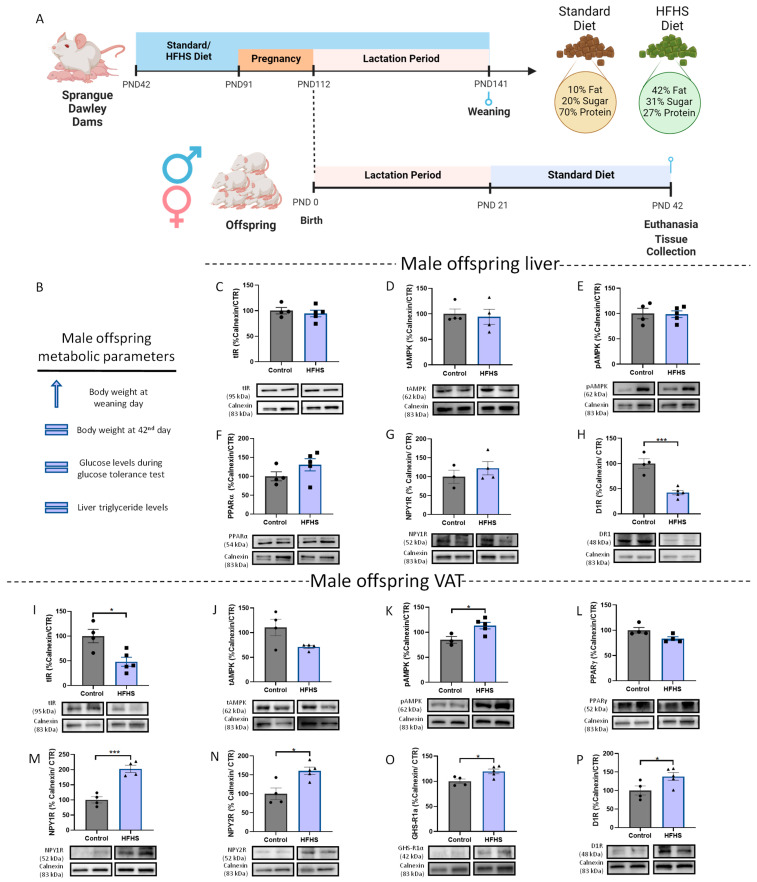
Maternal obesity increases both lipogenic and catabolic mechanisms in WAT from male offspring at PND 42. Maternal DIO experimental design (Created with BioRender.com, accessed on 25 January 2023) (**A**). Summary of metabolic parameters of male offspring after the data from [50] (**B**). Total liver IR (**C**), total AMPK (**D**), p-AMPK (**E**), PPARα (**F**), and NPY1R (**G**) levels remain similar to the control, while D1R levels are reduced in male offspring (**H**). Maternal obesity reduces total IR levels (**I**) in the WAT of male offspring. Total levels of AMPK (**J**) and PPARγ (**L**) were maintained in WAT, while D1R (**P**) AMPK phosphorylation (**K**), NPY1R (**M**), NPY2R (**N**), and ghrelin receptor (GHS-R1α) (**O**) levels increase in WAT of male offspring. Representative images of Western blot proteins of interest and loading controls (Calnexin) are shown at the bottom. Control: 42-day-old Sprague-Dawley males from dams fed a standard diet; HFHS: male offspring with 42-day-old Sprague-Dawley dams fed an HFHS diet. Bars represent mean ± SEM of 3, 4, or 5 animals per group, and unpaired t-tests were performed to compare the groups. * *p* < 0.05; *** *p* < 0.001.

**Figure 2 nutrients-15-01281-f002:**
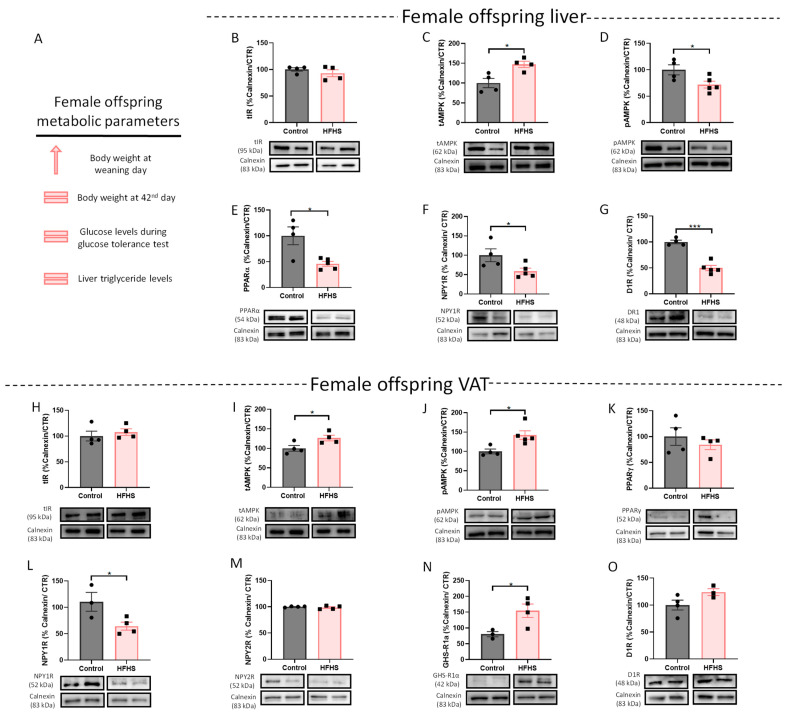
Effects of maternal DIO in liver and VAT of female offspring at PND 42. Summary of metabolic parameters of female offspring, after the date from [49] (**A**). Total liver IR levels were maintained (**B**), while PPARα (**E**), NPY1R (**F**), and D1R (**G**) levels are reduced in the liver of female offspring. Total AMPK levels increases (**C**) while AMPK phosphorylation decreased (**D**) in the liver of female rats from DIO dams. On WAT, total IR (**H**), PPARγ (**K**), NPY2R (**M**) and D1R (**O**) levels remain similar to the control. Maternal HFHS diet increases both total (**I**) and phosphorylated (**J**) forms of AMPK in female offspring. NPY1R levels were decreased (**L**) while GHS-R1α levels increased (**N**) in VAT of female offspring of DIO dams. Representative images of Western blot proteins of interest and loading controls (Calnexin) are shown at the bottom. Control—42-day-old Sprague-Dawley females from dams fed a standard diet; HFHS—female offspring with 42-day-old of Sprague-Dawley dams fed an HFHS diet. p-AMPK and D1R in liver samples were detected in the same membrane, as were PPARγ and NPY2R in VAT samples, which had the same loading control. Bars represent mean ± SEM of 3, 4, or 5 animals per group, and unpaired *t*-tests were performed to compare the groups. * *p* < 0.05; *** *p* < 0.001.

**Figure 3 nutrients-15-01281-f003:**
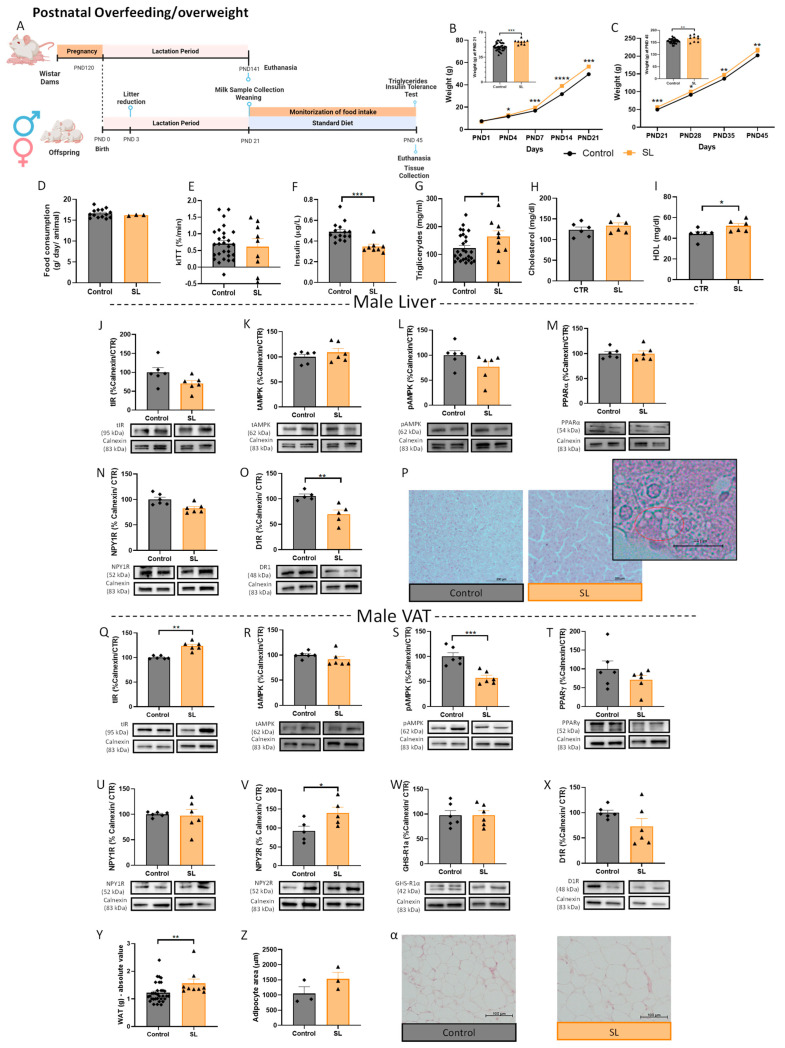
Postnatal overfeeding (SL) enhances VAT adipogenesis regulators and fat mass in male offspring. Postnatal overfeeding-induced early life obesity experimental design (created with BioRender.com, accessed on 25 January 2023) (**A**). Weight gain curves during the first 21 PND (weaning day) (**B**) and between PND 21 to PND 45 (period after weaning) (**C**). Postnatal overfeeding did not alter food intake in SL male animals after weaning (n = average per cage) (**D**). The kITT of overfed male animals on PND 45 (**E**). Postnatal overfeeding reduced plasma insulin levels (**F**) and increased plasma triglycerides (**G**) and HDL (**I**) in male SL animals without altering cholesterol plasma levels (**H**). Total IR (**J**), total AMPK (**K**), p-AMPK (**L**), PPARα (**M**), and NPY1R (**N**) levels remained similar to the control while D1R levels were reduced (**O**) in the liver of male overfed animals. Hematoxylin–eosin staining (100×) of liver from overfed male rats (**P**). Early life obesity increased total IR (**Q**) and NPY2R (**V**) levels in visceral AT, without affecting PPARγ (**T**), NPY1R (**U**), GHS-R1α (**W**), and D1R (**X**) levels in male SL animals. Overfeeding induced lower AMPK phosphorylation (**S**) but maintained total AMPK levels (**R**) in male SL animals. The absolute value of perigonal fat mass (mg) at PND 45 (**Y**), adipocytes area (**Z**), and representative images of perigonadal AT stained with hematoxylin–eosin (100×) (**α**). Control—45-day-old Wistar males from normal litters; SL—45-day-old Wistar males from SLs. IR and AMPK in liver samples were marked in the same membrane as well as IR and p-AMPK in VAT samples, having the same loading control. Bars represent the mean ± SEM of 35 (or 6 in WB data or 3 in adipocyte area) animals in the control and 9 (or 6 in WB data or 3 in adipocyte area) in the SL group, and unpaired t-tests were conducted to compare the groups. * *p* < 0.05; ** *p* < 0.01; *** *p* < 0.001.

**Figure 4 nutrients-15-01281-f004:**
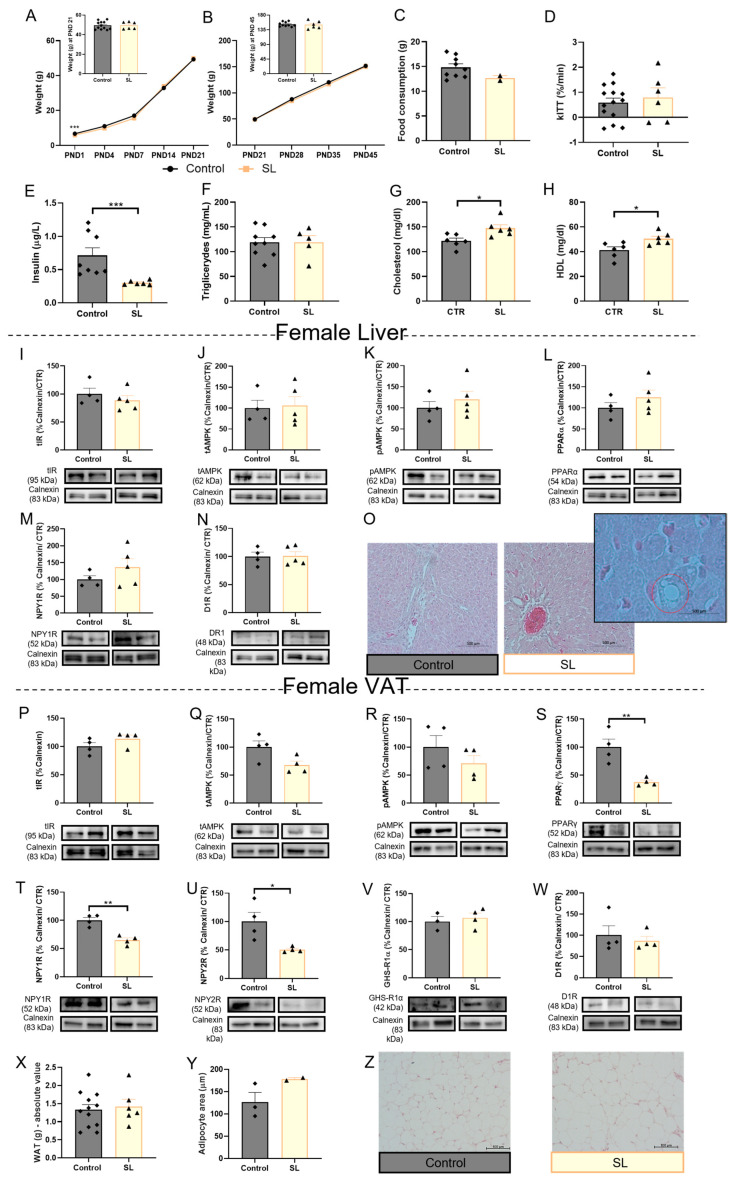
Adipogenic and lipogenic regulators were reduced in VAT female overfed rats. Weight gain curves during the first PND 21 (weaning day) (**A**) and between PND 21 to PND 45 (period after weaning) (**B**). Postnatal overfeeding did not alter food intake in obese female animals after weaning (**C**). The kITT of female overfed animals on PND 45 (**D**). Postnatal overfeeding induced by SL reduced plasma insulin levels (**E**), maintained plasma triglycerides (**F**), and increased both cholesterol (**G**) and HDL (**H**) levels. Total IR (**I**), total AMPK (**J**), p-AMPK (**K**), PPARα (**L**), NPY1R (**M**), D1R (**N**) levels remained similar to the control in the liver of SL females. Representative liver images stained with hematoxylin–eosin (100×) (**O**). Overfeeding during lactation did not affect total IR (**P**) total AMPK (**Q**), p-AMPK (**R**), GHS-R1α (**V**), and D1R (**W**) levels while impairing adipogenesis and lipogenesis-associated mechanisms NPY1R (**T**), NPY2R (**U**), and PPARγ (**S**) in VAT. The absolute value of fat mass (mg) at PND 45 (**X**), adipocytes area (**Y**), and representative images of perigonadal AT stained with hematoxylin–eosin (100×) (**Z**). Control—45-day-old Wistar females from normal litters; SL—45-day-old Wistar females from SLs. AMPK and PPARα, IR, and D1R in liver samples were marked in the same membrane as well as AMPK and NPY1R in VAT samples. Bars represent the mean ± SEM of 12 (or 4 in WB data or 3 in adipocyte area) animals in the control and 9 (or 4/5 in WB data or 2 in adipocyte area) in the SL group, and unpaired t-tests were conducted to compare among the groups. * *p* < 0.05; ** *p* < 0.01; *** *p* < 0.001.

**Figure 5 nutrients-15-01281-f005:**
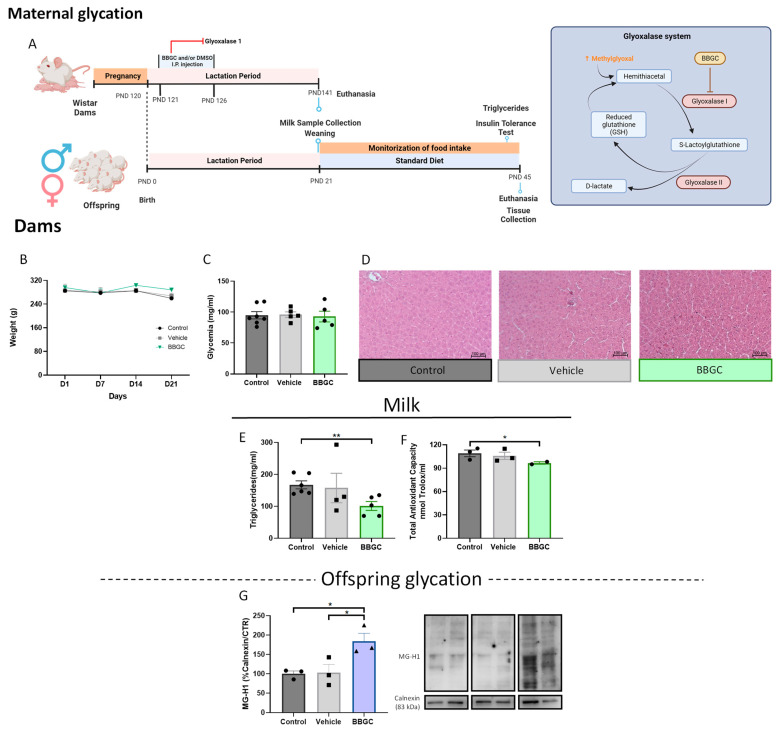
Maternal glycation alters breastmilk composition: Maternal glycation experimental design (created with BioRender.com, accessed on 25 January 2023) (**A**). Maternal weight gain curves during the first 21 days postpartum (**B**). Fasting glycemia of dams on the 21st postpartum day (**C**). Representative liver images from dams stained with hematoxylin–eosin staining (100×) (**D**). Glycation induced by BBGC reduced triglyceride levels (**E**) and total antioxidant capacity (**F**) content in breastmilk. Exposure to maternal glycation during lactation increases MG-H1 levels in VAT from male offspring (**G**). Control: Wistar control dams (or their offspring in **G**); vehicle: Wistar dams treated with DMSO (or their offspring in **G**); BBGC: Wistar dams treated with BBGC (5 mg/kg) through i.p. (or their offspring in **G**). Bars represent mean ± SEM of 2–5 dams per group or 3 rat offspring, and the Kruskal–Wallis test or one-way ANOVA was conducted to compare the groups. * *p* < 0.05; ** *p* < 0.01.

**Figure 6 nutrients-15-01281-f006:**
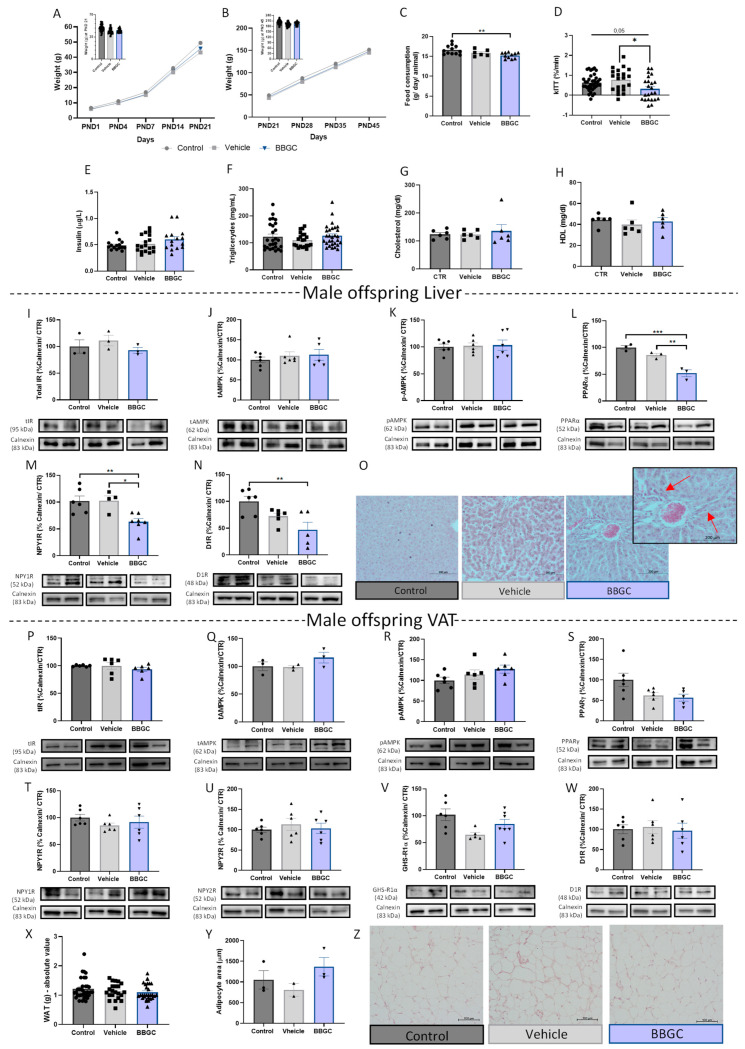
Maternal glycation reduced insulin sensitivity and liver NPY and D1R levels, not affecting VAT energy balance mechanisms in lean male offspring. Weight gain curves during the first PND 21 (weaning day) (**A**). Weight gain curves between PND 21 to PND 45 (period after weaning) (**B**). Maternal glycation reduced food intake in male offspring after weaning (**C**). Maternal glycation reduced the kITT (**D**) without affecting plasma insulin levels (**E**). Plasma triglyceride levels (**F**), plasma cholesterol levels (**G**) and plasma HDL levels (**H**) do not change. Offspring exposed to glycotoxins during lactation presented lower levels of PPARα (**L**), NPY1R (**M**), and D1R (**N**) while total IR (**I**), total AMPK (**J**), and p-AMPK (**K**) levels were conserved. Representative image of inflammatory infiltration in the liver of male offspring exposed to maternal glycation (**O**) (100×). Levels of total IR (**P**), total AMPK (**Q**), p-AMPK (**R**), PPARγ (**S**), NPY1R (**T**), NPY2R (**U**), GHS-R1α (**V**), and D1R (**W**) in VAT of male offspring submitted to glycotoxins exposure during lactation. The absolute value of fat mass (mg) at PND 45 (**X**), adipocytes area (**Y**), and representative images of perigonadal AT stained with hematoxylin–eosin (100×) (**Z**). Representative images of Western blot proteins of interest and loading controls (Calnexin) are shown at the bottom. Control: male offspring of control dams; vehicle: male offspring of dams treated with DMSO; BBGC- Wistar male offspring of dams treated with BBGC (5 mg/kg) through IP. Bars represent the mean ± SEM of 35 animals (or 6 in WB data or 3 in adipocyte area) in control group, 22 (or 6 in WB data or 2 in adipocyte area) in vehicle group, and 29 in BBGC group (or 6 in WB data or 3 in adipocyte area) and one-way ANOVAs were performed to compare the groups. * *p* < 0.05; ** *p* < 0.01; *** *p* < 0.001.

**Figure 7 nutrients-15-01281-f007:**
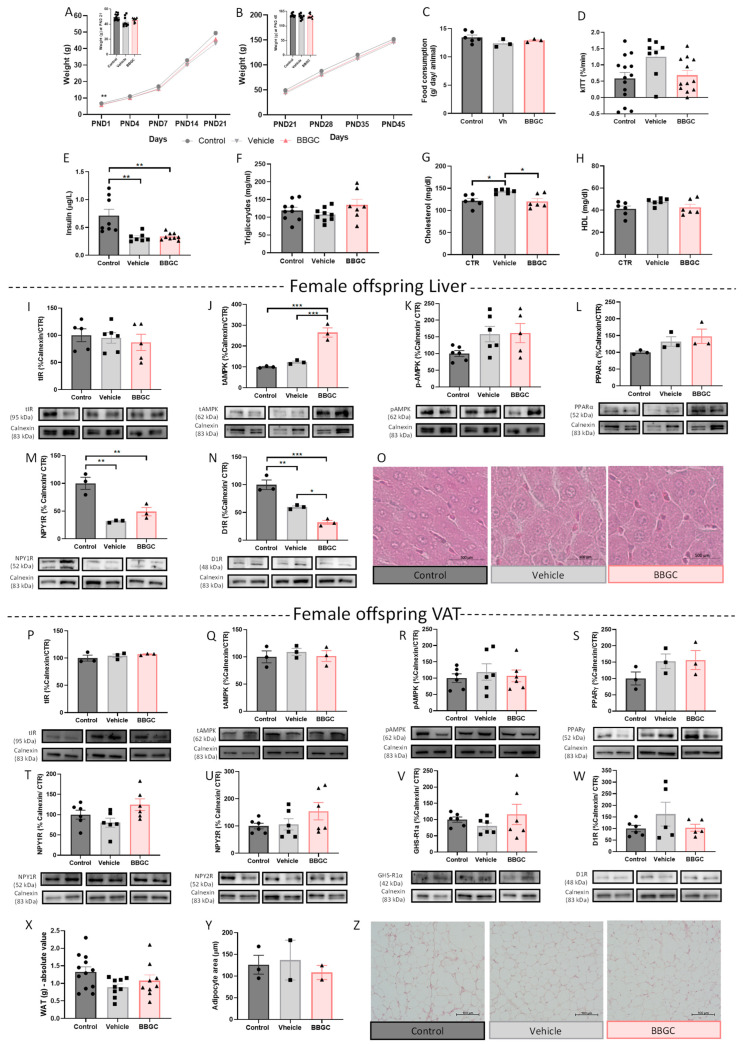
Maternal glycation does not affect metabolic profile and VAT energy balance in BBGC female offspring. Weight gain curves until 21 PND (weaning day) (**A**) and between PND 21 to PND 45 of female offspring (**B**). Maternal glycation did not alter food intake after weaning (**C**), the kITT (**D**), and plasma levels of insulin (**E**), triglyceride levels (**F**), total cholesterol (**G**), and HDL-cholesterol (**H**) in female offspring exposed to maternal glycotoxins. Maternal glycation increased AMPK levels (**J**) in the liver of female offspring without affecting AMPK phosphorylation (**K**), total IR (**I**), and PPARα levels (**L**). Both vehicle and BBGC treatment in dams decreased NPY1R (**M**) and D1R (**N**) levels in the liver of female offspring. Representative images of liver from female offspring stained with hematoxylin–eosin staining (100×) (**O**). Levels of total IR (**P**), total AMPK (**Q**), p-AMPK (**R**), PPARγ (**S**), NPY1R (**T**), NPY2R (**U**), GHS-R1α (**V**), and D1R (**W**) in WAT of female offspring submitted to maternal glycation. The absolute value of fat mass (mg) at PND 45 (**X**), adipocytes area (**Y**), and representative images of perigonadal AT stained with hematoxylin–eosin (100×) (**Z**). Representative images of Western blot proteins of interest and loading controls (Calnexin) are shown at the bottom. Control: female offspring of control dams; vehicle: female offspring of dams treated with DMSO; BBGC: Wistar female offspring of dams treated with BBGC. AMPK and PPARα in liver samples were marked in the same membrane. Bars represent the mean ± SEM of 12 (or 3/6 in WB data or 3 in adipocyte area) animals in control group, 9 (or 3/6 in WB data or 2 in adipocyte area) in vehicle group, and 9 (or 3/6 in WB data or 2 in adipocyte area) in BBGC group, and one-way ANOVAs were conducted to compare the groups. * *p* < 0.05; ** *p* < 0.01; *** *p* < 0.001.

**Figure 8 nutrients-15-01281-f008:**
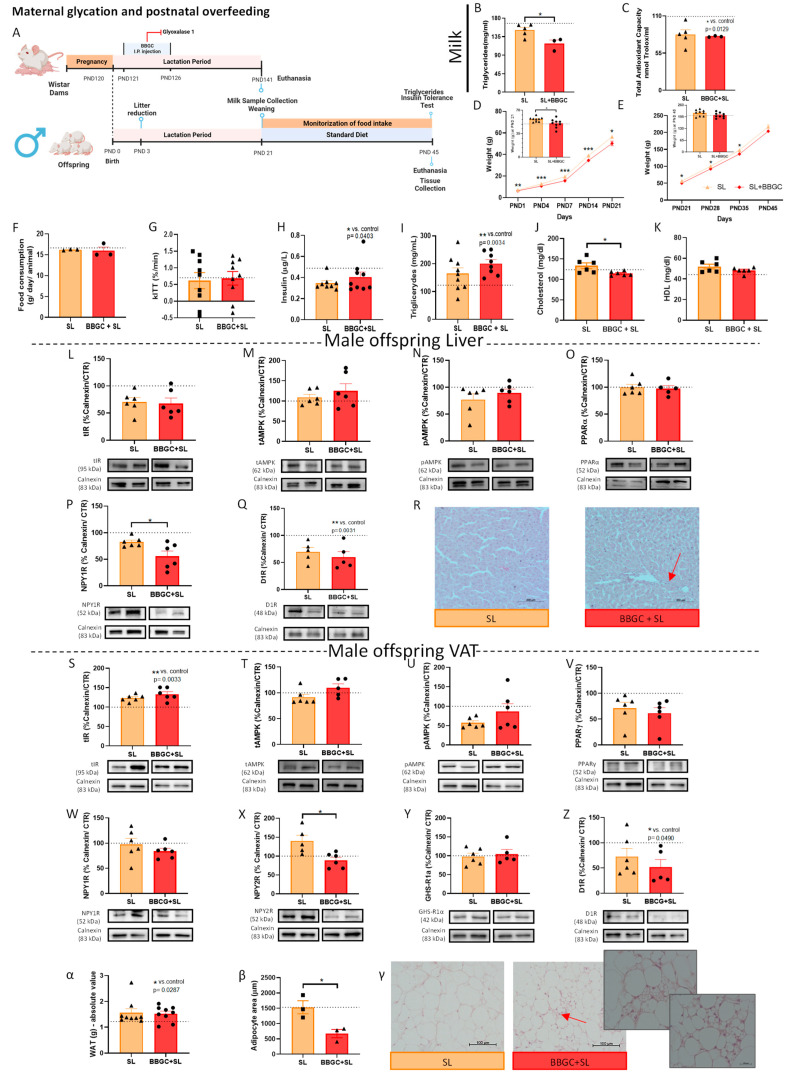
Maternal glycation impairs energy storage preventing weight gain in male offspring overfed induced by SL. Maternal glycation on SL-induced overfed rats experimental design (created with BioRender.com, accessed on 25 January 2023) (**A**). The triglyceride levels were decreased in breastmilk of BBGC dams (**B**). The total antioxidant capacity in breastmilk was affected by both the SL process and BBGC treatment (**C**). Weight gain curves during the first PND 21 (weaning day) (**D**) and between PND 21 to PND 45 (period after weaning) (**E**). Postnatal overfeeding did not alter food intake in male SL offspring exposed to maternal glycation after weaning (**F**). The kITT of SL offspring from dams treated with BBGC on the PND 45 (**G**). Plasma insulin levels (**H**), plasma triglyceride levels (**I**), plasma cholesterol levels (**J**), and plasma HDL levels (**K**) of overfed offspring exposed to maternal glycation. The levels of total IR (**L**), total AMPK (**M**), p-AMPK (**N**), PPARα (**O**), and D1R (**Q**) in the liver of overfed offspring exposed to maternal glycation remained similar to the animals with early-life obesity. NPY1R levels (**P**) in the liver of overfed animals exposed to glycotoxins during lactation were decreased compared to the postnatal overfed ones. Representative images of liver stained with hematoxylin–eosin (100×) (**R**). Compared to obese animals, the levels of total IR (**S**), total AMPK (**T**), p-AMPK (**U**), PPARγ (**V**), NPY1R (**W**), GHS-R1α (**Y**), and D1R (**Z**) were maintained in WAT from postnatal overfed offspring exposed to maternal glycation, while NPY2R (**X**) levels and adipocytes area (**β**) were decreased. The absolute value of fat mass (mg) at PND 45 (**α**) and representative images of perigonadal AT stained with hematoxylin–eosin (100×) (**γ**). SL: 45-day-old Wistar males from SLs; BBGC +SL: 45-day-old male obese offspring of dams treated with BBGC. Bars represent the mean ± SEM of 9 (or 6 in WB data or 3 in adipocyte area) animals in the SL group and 9 (or 6 in WB data or 3 in adipocyte area) animals in the BBGC + SL group, and unpaired t-tests were performed to compare the groups. * *p* < 0.05. Dashed lines represent the value of control animals.

**Figure 9 nutrients-15-01281-f009:**
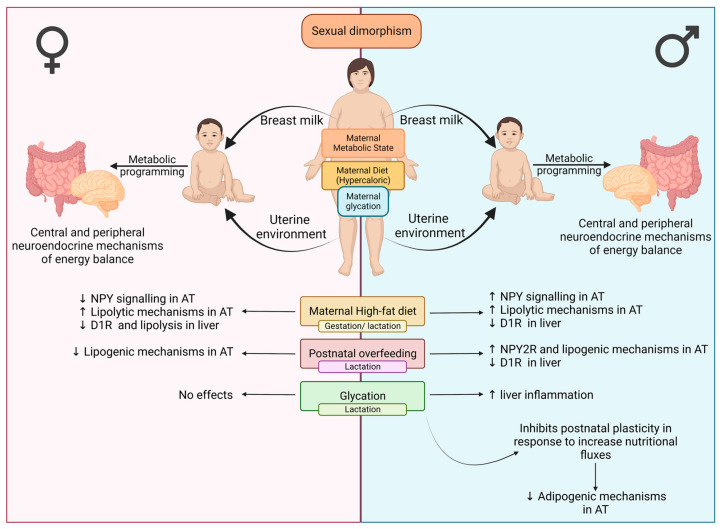
Hypothetical effects of obesogenic environments (hypercaloric diet, postnatal overfeeding, and maternal glycation) on human energy balance mechanisms depending on sexual dysmorphism observed in rat models (created with BioRender.com, accessed on 25 January 2023).

## Data Availability

The datasets generated during and/or analyzed during the current study are available from the corresponding author upon reasonable request.

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
