# Peer review of "Exposure to Obesogenic Environments during Perinatal Development Modulates Offspring Energy Balance Pathways in Adipose Tissue and Liver of Rodent Models"

_nutrients, 2023, doi:10.3390/nu15051281_

Round 1

Reviewer 1 Report

This study investigated the impact of obesogenic environments in perinatal time on the offspring energy balance pathways in adipose tissue and liver. Overall, the study was well-performed and presented. Some revisions are suggested.

The original bands of Western blotting should be supplied in the supplementary data.

All the figures created by Biorender are too dim, and take care of the copyright of the cartoons.

In Figures, why different animal numbers were chosen to evaluate protein expression? Is there some big variation in the results?

Line 33, exhibited higher VAT NPY2R levels > exhibited higher NPY2R levels in VAT.

Lines 135-138, NL= 12, SL=9, with/without space to be consistent.

Line 143, 60µL > 60 µL.

List a title for Supplementary Table.

Author Response

Dear Editor,

We thank the reviewers for the time invested in the revision process and for the constructive comments that resulted in improvements in the manuscript. In this document, we respond, point by point, to the concerns and questions raised. The alterations made in the manuscript following the reviewers’ questions are in track changes mode in the text (and detailed in each reply) so that they can be easily found.

We believe we have made all the required alterations and hope that they meet your expectations.

Response to Reviewer 1 Comments

Point 1: The original bands of Western blotting should be supplied in the supplementary data.

Response 1: In the first submission we added the original bands to supplementary data to not publish. However, may have happened some error with the upload. We apologize for that, and we are resubmitting the document now.

Point 2: All the figures created by Biorender are too dim, and take care of the copyright of the cartoons.

Response 2: We cannot do anything regarding the tone of Biorender figures. We downloaded the pictures from biorender.com with high quality together with the copyright certification. We increased font size and added the certification of the copyright to the supplementary data to not publish.

Point 3: In Figures, why different animal numbers were chosen to evaluate protein expression? Is there some big variation in the results?

Response 3: The number of samples available in each model was different, due to the number of pups per dam. In all the parameters measured in vivo, all the animals available were tested, while a smaller number of samples was selected for biochemical analyses. In all the measurements performed, at least one sample was selected from each dam.

  • Regarding the maternal DIO model, we use 4 controls and 5 HFHS samples, all available from previous studies. In males, one of the samples did not run appropriately in some gels, so it was not, namely in the membranes marked with AMPK total, p-AMPK and PPARγ in VAT and PPARα in the liver. In the VAT of female HFHS, the number of available samples was even lower, limiting the number of samples to load in SDS gels. So, in the membranes marked with PPARγ, NPY2R and D1R there are only 8 samples: 4 controls and 4 HFHS.
  • The male SL samples were loaded with BBGC+SL samples. We loaded 2 gels for each protein of interest: 3 controls, 3 SL and 3 BBGC+SL per membrane. The membranes shown in supplementary data correspond to those shown in the paper. In females, we loaded the gels with 4 control and 5 SL samples, and 4 controls and 4 HFHS VAT samples, which is a number sufficient for the statistics.
  • In the maternal glycation model, we also load 2 gels for the almost interest proteins in both sexes. In each gel, 3 controls, 3 vehicles, and 3 BBGC samples were loaded. In males, we quantified 2 membranes for all proteins except for PPARα and IR total in the liver and AMPK total in VAT. In the female BBGC model, we quantify 6 bands of IR total, p-AMPK in membranes with liver samples and p-AMPK, NPY1R, NPY2R, GHS-R1α and D1R in membranes with VAT samples. The remaining proteins of interest were marked on 1 membrane (3 bands for each condition)

Point 4: Line 33, exhibited higher VAT NPY2R levels > exhibited higher NPY2R levels in VAT.

Response 4: We acknowledge the incorrect phase order, therefore, we reorder the sentence as you suggested (in track changes - line 34).

Point 5: Lines 135-138, NL= 12, SL=9, with/without space to be consistent.

Response 5: Thank you for noticing that. We put a space between the L and the equal symbol (in track changes - line 144-148).

Point 6: Line 143, 60µL > 60 µL.

Response 6:  Thank you for noticing that. We put a space between 60 and µL (in track changes - line 153).

Point 7: List a title for Supplementary Table.

Response 7: Thank you for the suggestion. We add a title to the supplementary table.

Reviewer 2 Report

The authors exposed rat offspring to obesogenic environments and analyzed markers of energy balance in adipose tissue and livers. Even though the subject is interesting, the analytical methods are limited to western blotting and the results need to be validated by additional quantitative methods like ELISA.

Title

The title does not state that the data were obtained in a rat model. This is an essential information and must be added.

Abstract

Likewise, the kind of model should be made very clear from the beginning.

Introduction

p.2, l 80: „NPY and acyl ghrelin levels… contributing to adiposity and insulin sensitivity“ I would rather assume „insulin resistance“

p. 3, 100-105: The introduction lacks background information on the effects of maternal glycation and does not conclusively explain the hypotheses made by the authors.

The authors have divided the results of 1 study into several related manuscripts. Why? The introduction should give an outline on the previously published data of the study. Which parts of the data have already been published – and what do the additional findings add to it? Why are they published separately?

Methods

2.1.1. The authors do not describe the model of DIO, but refer to the previously published parts of the study in related manuscrips. Nonetheless, a brief description of the experimental design is necessary including species and number of animals.

2.1.2. Why did the authors use Wistar rats in this model? The model of DIO was performed in Sprague-Dawley rats and it would be reasonable to stick to one strain to obtain comparable results. The authors should report on the number of dams per group.

Postnatal overfeeding and maternal glycosylation models: How were the models validated? Give references.

Results

The quantification of metabolic effects is only based on western blots. In many cases, a selection of few samples (and not the whole group) was examined. To perform statistics, the whole group should be considered (exceptions must be justified), and at least 3 litters. Otherwise, I recommend against a statistical analysis.

Western blotting is regarded as semi-quantitative and the results should be validated by other methods like ELISA.

Mean liver weights should be reported in all groups.

In addition to insulin and triglycerides, blood samples should be analyzed for glucose, liver enzymes (ALT, AST), cholesterol, HDL, and LDL.

In the maternal glycation model, the course of dam body weight under treatment should be reported.

Figures

The resolution of histologic stainings, immunoblots, and experimental design schemes is not sufficient for evaluation.

Figure legends: The indicated numbers of animals/samples per group does not correspond to the number of dots in the single graphs. Please correct. How many litters are represented per group? Some of the graphs report only 2-3 samples, but the original group was much bigger (like n=9).

Figure 1: The previously published study data [references 46,47] report on n=6 per group. Why does the figure only report n=3-5 per group?

Figure 3: The number of animals per group is very different: n=35 in the control group and only n=9 in the SL group – Why? This is an issue for statistical validity. Furthermore, the number of animals per group in some of the graphs is much lower (n=3-6). The rationale behind data selection must be outlined. How many litters were represented per group?

Figure 4, 5, 6, 7, 8: Same issue as in figure 3.

All parts of the manuscript including every figure legend etc. should state that the manuscript is dealing with rat data, not with human-derived data. Figure 9 should be labelled with „hypothetical effects of obesogenic environments in humans“

Discussion

The authors should discuss the limitations of their study and potential sources of bias.

Non-published data:

-          The molecular markers should be shown on the same membrane, not cut.

-          The labelling of the control and treatment groups is not clear and some of the bands are not labelled. The lines should be placed more precisely.

Author Response

Dear Editor,

We thank the reviewers for the time invested in the revision process and for the constructive comments that resulted in improvements in the manuscript. In this document, we respond, point by point, to the concerns and questions raised. The alterations made in the manuscript following the reviewers’ questions are in track changes mode in the text (and detailed in each reply) so that they can be easily found.

We believe we have made all the required alterations and hope that they meet your expectations.

Response to Reviewer 2 Comments

Point 1: The title does not state that the data were obtained in a rat model. This is an essential information and must be added.

Response 1: We acknowledge the lack of that information. We change the title to improve the clarity of the study (in track changes).

Point 2: Abstract: Likewise, the kind of model should be made very clear from the beginning.

Response 2: Thank you for your suggestion, we add that information to the abstract (in track changes - line 28).

Point 3: p.2, l 80: „NPY and acyl ghrelin levels… contributing to adiposity and insulin sensitivity“ I would rather assume „insulin resistance“.

Response 3: There was a mistake, NPY and acyl ghrelin reduce the insulin sensitivity, not induced insulin resistance. We acknowledge the error, and we corrected the sentence (in track changes - line 84).

Point 4: p. 3, 100-105: The introduction lacks background information on the effects of maternal glycation and does not conclusively explain the hypotheses made by the authors.

Response 4: We acknowledge the lack of background relating to glycation. We add more studies to support the aim of this work (track change - lines 58 to 64).

Point 5: The authors have divided the results of 1 study into several related manuscripts. Why? The introduction should give an outline on the previously published data of the study. Which parts of the data have already been published – and what do the additional findings add to it? Why are they published separately?

Response 5: We initially designed the project with 3 animal models: postnatal overfeeding (SL), maternal glycation, and maternal glycation combined with postnatal overfeeding. Then, we hypothesized whether other models of postnatal overweight would have similar mechanisms as those observed in the SL model. Thus, we used the samples from another study already published (maternal DIO), allowing the comparison between several obesogenic environments during different perinatal periods (gestational and lactation). The principles of the 3R defend Replacement, Reduction, and Refinement, promoting the usage of different tissues from the same animals. This would allow the use of a smaller number of animals and simultaneous more scientific research. Data already published regarding maternal DIO (HFHS diet) describe metabolic parameters (body weight and glucose levels during GTT), hepatic fat accumulation and mitochondrial activity in the liver of offspring exposed to maternal DIO. Furthermore, the effect of the study also describes the maternal exercise on preventing mitochondrial activity. Here, we only use VAT and liver to study the energy balance mechanisms in animals exposed to maternal DIO. Thus, studies already published have different purposes, and that’s the reason for different research articles.

Point 6: T 2.1.1. The authors do not describe the model of DIO, but refer to the previously published parts of the study in related manuscripts. Nonetheless, a brief description of the experimental design is necessary including species and the number of animals.

Response 6: Thank you for the suggestion, we add a brief description of the maternal DIO experimental design which improve the read of the manuscript (track changes – line 113 to 123).

Point 7: 2.1.2. Why did the authors use Wistar rats in this model? The model of DIO was performed in Sprague-Dawley rats and it would be reasonable to stick to one strain to obtain comparable results. The authors should report on the number of dams per group.

Response 7: As mentioned in point 5, first we design a project including three animal models (postnatal overfeeding, maternal glycation and maternal glycation combined with postnatal overfeeding). The data regarding Sprague-Dawley rats were already published, and we decided to include this model in our work, applying the principles of the 3R. Despite the different strains, all models are rats, allowing the comparison between the obesogenic models which is the main objective.

The number of dams corresponds to the number of litters, so we added that information accompanying the number of animals per group that was already described in the manuscript (track changes – lines 120 to 123, 147, 153, and  161).

Point 8: Postnatal overfeeding and maternal glycosylation models: How were the models validated? Give references.

Response 8: The protocol used to induce postnatal overfeeding consists of the reduction of the adjustment by three pups per litter. This way, the dams during the first three days adjust the milk production to normal litter, and on the 3rd day, the litter is reduced to three and the remaining pups are exposed to higher availability of breastmilk, which leads to higher food intake. The SL model was already described as a postnatal overfeeding model as well as postnatal overweight [1–3]. Indeed this was observed in both females and males from SL at 5-month-old [1].

Diabetes and insulin resistance are highly related to AGEs. Moreover, diabetic patients present not only higher concentrations of methylglyoxal, an AGEs precursor, but also greater activity of glyoxalase I in first stages and lower activity later, suggesting that chronic exposure to methylglyoxal potentiates glyoxalase system activity, which is later lost [4]. Some studies reported an increase in methylglyoxal after BBGC treatment in human and mouse cells [5,6]. In cancer, the inhibitor of glyoxalase I – BBGC - is used as a co-treatment at supraphysiological doses (50, 100 and 200 mg/kg) [7]. In our pilot study (data not shown), we used 2 doses ( 5 and 10 mg/kg) to test toxicity, without histological findings for both of them. As we intended to use a physiologic dose, we administrated 5 mg/kg, which resulted in increased AGEs in the VAT of the offspring, as now shown in figure 5. For the first time to our knowledge, we induced a model of maternal glycation using BBGC. Nevertheless, the administration of methylglyoxal by gavage in dams was already used in order to induce maternal glycation, affecting offspring metabolism [8]. However, since there is a direct administration of an AGEs precursor, translation to human situations is less realistic and may result in supraphysiological doses. To avoid this, we decided to inhibit an enzyme responsible for metabolizing methylglyoxal. In this way, methylglyoxal is accumulated naturally and not by exogenous administration. We semi-quantified VAT offspring levels N_-(5-hydro-5-methyl-4-imidazolon-2-yl)-ornithine (MG-H1), an AGE derived from methylglyoxal [9]. Therefore, we can confirm the accumulation of AGEs in offspring when exposed to maternal glycation. We add this blot to figure 5 in the manuscript and also a brief explanation for MG-H1 formation (track changes line 373). Furthermore, in another article already submitted to another journal and containing data from the same animal models, we demonstrated the accumulation of MG-H1 and argpyrimidine (Arg-p), other methylglyoxal-derived AGEs, in brain areas (cortex and hippocampus).

Point 9: The quantification of metabolic effects is only based on western blots. In many cases, a selection of few samples (and not the whole group) was examined. To perform statistics, the whole group should be considered (exceptions must be justified), and at least 3 litters. Otherwise, I recommend against a statistical analysis.

Response 9: Despite we have a large considerable number of samples regarding metabolic parameters, in WB we used mostly 5-6 animals/group, and at least one for all the litters studied. We used SDS gels with 9 wells. Thus, the amount of gels and membranes need to perform WB for all the samples that we obtain was incredibly high. Furthermore, we do not have the same number of samples in all groups to load the gels with the different conditions properly since in each membrane we need at least 1 sample from each group.

  • Regarding the maternal DIO model, we use 4 controls and 5 HFHS samples (1 membrane for each protein of interest). We used samples from different litters in all western blots.
  • The male SL samples were loaded with BBGC+SL samples. We loaded 2 gels for each interest protein: 3 controls (from different litters), 3 SL (from different litters) and 3 BBGC+SL (from different litters) per membrane. In female we load the gels with 4 control (from different litters) and 5 SL samples (samples from 3 litters). In VAT of female SL, we included 4 controls and 4 HFHS samples to the statistic as showed in original bands of western blotting.
  • In maternal glycation model, we also load 2 gels for the almost interest proteins in both sexes. In each gel 3 controls (from different litters), 3 vehicles (from different litters) and 3 BBGC samples (from different litters) were load. The remaining proteins of interest were marked in 1 membrane (3 bands for each condition).

In conclusion, at least samples from 3 different litters were used in each western blot.

Point 10: Western blotting is regarded as semi-quantitative and the results should be validated by other methods like ELISA.

Response 10: To measure insulin plasma levels, we performed an ELISA KIT. With this scientific work we intended to understand the alterations at cellular level in several obesogenic conditions. In tissues, ELISA would not provide quantitative data, because all the data are also expressed in relation to the amount of protein in the extracted sample. Thus, in tissue samples, ELISA is also a semi-quantitative technique.

Point 11: Mean liver weights should be reported in all groups.

Response 11: Our figures are very filed and since the liver weights were not changed, we choose to eliminate this data. In the resubmission of the manuscript, we added the information regarding liver weights (in track changes – line 297, 327, 397, 429 and 479) and the graphs to supplementary data.

Thank you for your suggestion.

Point 12: In addition to insulin and triglycerides, blood samples should be analyzed for glucose, liver enzymes (ALT, AST), cholesterol, HDL, and LDL.

Response 12: Regarding glucose levels, we performed an insulin tolerance test (ITT). In the manuscript we showed the decay of the glucose rate during the insulin tolerance test per minute (KITT) which is the best was to analyze insulin sensitivity. The point 0 of ITT corresponds to the mean glucose levels under fasting conditions on the 45th day. The following graphs correspond to ITT results and glucose plasma levels.

Regarding the liver enzymes (ALT, AST), since the livers did not effectively present steatosis, we decided not to measure.

We already had the data of cholesterol and HDL levels in plasma from SL, maternal glycation, and SL + maternal glycation models. We added to the figures (3, 4, 6, 7 and 8). Since we only have 10 days to respond we do not have LDL data.

Point 13: In the maternal glycation model, the course of dam body weight under treatment should be reported.

Response 13: The figure 5B is the body weight of dams from day 1 to day 21. The 6 days of BBGC treatment started on day 1. Therefore, in the manuscript, the course of the dam body weight under treatment was already shown.

Point 14: The resolution of histologic stainings, immunoblots, and experimental design schemes is not sufficient for evaluation.

Response 14: Regarding histologic staining we added the images to supplementary data with greater dimensions. As the manuscript has many filed figures, put the histologic images with greater dimensions would make the manuscript too long. Regarding immunoblots and experimental design schemes, we believe they have the proper resolution. It is possible that during the submission the quality of the figures and its resolution suffer some problem. We apologize and hope that now the quality and resolution of the figures are improved.

Point 15: Figure legends: The indicated numbers of animals/samples per group does not correspond to the number of dots in the single graphs. Please correct. How many litters are represented per group? Some of the graphs report only 2-3 samples, but the original group was much bigger (like n=9).

Response 15: We acknowledge that the number of animals were in some cases misfit. We corrected that in all the titles of the figures. In all the data analyzed, at least 3 litters were represented even in western blot as explained in point 9. As explained in point 9, western blots were performed with n=5-5, not all the samples. In graphs with adipocyte area, we analyzed 2-3 animals/group. However, we took data from at least 10 photos per animal, and each point in the graphs represents the mean of several VAT areas of each animal.

Point 16: Figure 1: The previously published study data [references 46,47] report on n=6 per group. Why does the figure only report n=3-5 per group?

Response 16: The work of the previously published study data was already completed, and some samples were ended. Thus, we choose to analyze 4 control and 5 HFHS to complete the 9 spaces of the genls. In some specific cases, we had a problem in the run of the gels and those samples were eliminated, as shown in the original gels.

Point 17: Figure 3: The number of animals per group is very different: n=35 in the control group and only n=9 in the SL group – Why? This is an issue for statistical validity. Furthermore, the number of animals per group in some of the graphs is much lower (n=3-6). The rationale behind data selection must be outlined. How many litters were represented per group?

Response 17: The control litters were normalized for n=9 pups, while the SL litters were reduced to n=3 pups/litter, which is the reason for such difference. The number of rats results from at least 3 litters/ experimental group. The number of litters was added to the manuscript as required in point 7.

Point 18: Figure 4, 5, 6, 7, 8: Same issue as in figure 3.

Response 18: In point 17 we already respond to this issue with the number of samples per group.

Point 19: All parts of the manuscript including every figure legend etc. should state that the manuscript is dealing with rat data, not with human-derived data. Figure 9 should be labelled with „hypothetical effects of obesogenic environments in humans“

Response 19: Thank you for your suggestion, we altered the title of figure 9 to address that question.

Point 20: The authors should discuss the limitations of their study and potential sources of bias.

Response 20: We believe that most of the limitations were already mentioned in the discussion and conclusion. However, we further explored some points.

Point 21: The molecular markers should be shown on the same membrane, not cut.

Response 21: Our software does not show the molecular marker in colors. So, we added to each membrane the molecular marker in color. But the membrane has molecular marker in black and white:

Some membranes do not have a molecular marker because the person who took them did not save the original image. We apologize for that, but we cannot access the original file.

Point 22: The labelling of the control and treatment groups is not clear and some of the bands are not labelled. The lines should be placed more precisely.

Response 22: We apologize for that. We added lines to the file with the original western blots to be clearer.

References:

[1]           Rodrigues VST, Moura EG, Peixoto TC, Soares PN, Lopes BP, Bertasso IM, et al. The model of litter size reduction induces long‐term disruption of the gut‐brain axis: An explanation for the hyperphagia of Wistar rats of both sexes. Physiological Reports 2022;10. https://doi.org/10.14814/phy2.15191.

[2]           Schellong K, Neumann U, Rancourt RC, Plagemann A. Increase of Long-Term ‘Diabesity’ Risk, Hyperphagia, and Altered Hypothalamic Neuropeptide Expression in Neonatally Overnourished ‘Small-For-Gestational-Age’ (SGA) Rats. PLoS ONE 2013;8:e78799. https://doi.org/10.1371/journal.pone.0078799.

[3]           Conceição EPS, Carvalho JC, Manhães AC, Guarda DS, Figueiredo MS, Quitete FT, et al. Effect of Early Overfeeding on Palatable Food Preference and Brain Dopaminergic Reward System at Adulthood: Role of Calcium Supplementation. J Neuroendocrinol 2016;28. https://doi.org/10.1111/jne.12380.

[4]           McLellan AC, Thornalley PJ, Benn J, Sonksen PH. Glyoxalase System in Clinical Diabetes Mellitus and Correlation with Diabetic Complications. Clinical Science 1994;87:21–9. https://doi.org/10.1042/cs0870021.

[5]           Nigro C, Raciti GA, Leone A, Fleming TH, Longo M, Prevenzano I, et al. Methylglyoxal impairs endothelial insulin sensitivity both in vitro and in vivo. Diabetologia 2014;57:1485–94. https://doi.org/10.1007/s00125-014-3243-7.

[6]           Kuhla B, Lüth H-J, Haferburg D, Weick M, Reichenbach A, Arendt T, et al. Pathological effects of glyoxalase I inhibition in SH-SY5Y neuroblastoma cells. J Neurosci Res 2006;83:1591–600. https://doi.org/10.1002/jnr.20838.

[7]           Kung Y, Wyatt C, Davies N. Antitumour Activity of S-p-bromobenzylglutathione Cyclopentyl Diester in. Vitro and in. Viwo. n.d.

[8]           Francisco FA, Barella LF, Silveira S da S, Saavedra LPJ, Prates KV, Alves VS, et al. Methylglyoxal treatment in lactating mothers leads to type 2 diabetes phenotype in male rat offspring at adulthood. Eur J Nutr 2018;57:477–86. https://doi.org/10.1007/s00394-016-1330-x.

[9]           Matafome P. Another Player in the Field: Involvement of Glycotoxins and Glycosative Stress in Insulin Secretion and Resistance. Diabetology 2020;1:24–36. https://doi.org/10.3390/diabetology1010004.
